# Breast Cancer Orbital Metastases: Clinical and Histopathological Characteristics, Imaging Features, and Disease-Related Survival in a Multicentric Retrospective Case Series

**DOI:** 10.3390/cancers17111875

**Published:** 2025-06-03

**Authors:** Sofia Peschiaroli, Adriana Iuliano, Giovanni Cuffaro, Francesco M. Quaranta Leoni, Tommaso Tartaglione, Monica Maria Pagliara, Maria Grazia Sammarco, Carmela Grazia Caputo, Angela Santoro, Matteo Barchitta, Vittoria Lanni, Diego Strianese, Gustavo Savino

**Affiliations:** 1Ocular Oncology Unit, Fondazione Policlinico Universitario A. Gemelli-IRCCS, Università Cattolica del Sacro Cuore, 00168 Rome, Italy; 2Department of Neurosciences, Reproductive and Odontostomatological Sciences, Division of Ophthalmology, School of Medicine, University of Naples “Federico II”, 80131 Naples, Italy; 3Orbital and Adnexal Service, Tiberia Hospital—GVM Care & Research, Via Emilio Praga 26, 00137 Rome, Italy; 4Department of Translational Medicine, University of Ferrara, 44121 Ferrara, Italy; 5Dipartimento di Diagnostica per Immagini, Fondazione Policlinico Universitario A. Gemelli-IRCCS, Università Cattolica del Sacro Cuore, 00168 Rome, Italy; 6Unità di Anatomia Patologica Generale, Dipartimento Scienze della Salute della Donna, del Bambino e di Sanità Pubblica, Istituto di Anatomia Patologica, Fondazione Policlinico Universitario A. Gemelli-IRCCS, Largo A. Gemelli 8, 00168 Rome, Italy

**Keywords:** orbital tumors, orbital metastases, breast carcinoma metastases, breast carcinoma orbital metastases, breast carcinoma lobular histotype

## Abstract

Orbital metastases from breast cancer are uncommon but represent a significant clinical challenge, often indicating advanced systemic disease. Their diagnosis and management can be complex due to the varied presentation and delayed onset following the primary tumor. This study aims to enhance understanding of the clinical, radiological, and histopathological characteristics of breast cancer orbital metastases, as well as their progression over time. By analyzing data from a multicenter case series, the authors seek to identify common features such as the most frequent tumor subtype, timing of orbital involvement, and patient outcomes. The findings provide valuable insights into the natural history and prognosis of orbital metastases, particularly in relation to histological type and survival. This research may contribute to earlier recognition, more informed clinical decision-making, and improved multidisciplinary care for patients affected by this manifestation of breast carcinoma.

## 1. Introduction

Orbital metastases from breast cancer (BC) represent a significant clinical challenge due to their prevalence and impact on patient prognosis, and they are the most common primary tumor leading to orbital metastases, accounting for approximately 30–60% of such cases. Up to 10% of patients affected by BC develop orbital metastases [1,2,3]. Typically, the presentation is unilateral, although a few bilateral cases have been described [4]. In up to 26% of cases, orbital metastases are observed as the primary manifestation of the disease, prior to the diagnosis of BC [5,6]. Metastatic disease is associated with an advanced stage of illness, and generally, orbital metastases occur with a latency of 3–6 years from the diagnosis of the primary tumor [7,8].

Ductal carcinoma (no special type—NST) is the most common histotype of primitive BC, representing about 80% of cases, but it is an uncommon cause of orbital metastases. Lobular carcinoma represents approximately 10% of BC and accounts for the majority of the tumors causing metastases to the orbit [9]. BC molecular profiling has a proven prognostic and predictive value [10,11], and there is clear evidence that some variants have the tendency to metastasize in specific anatomical districts. Biomarkers and molecular profile differences have also been reported between the primary tumor and metastasis [12,13].

The clinical value of the early identification of breast cancer is crucial for reducing the risk of metastatic spread and improving patient outcomes. Monitoring early metastatic spread is a critical aspect of managing breast cancer, as it helps detect distant metastases before they become symptomatic, potentially improving prognosis and allowing for earlier intervention. Computed tomography (CT) and magnetic resonance imaging (MRI) are pivotal in detecting and characterizing orbital metastases and assessing disease progression, which are fundamental for determining the most appropriate therapeutic strategies [14]. Orbital biopsy is essential for definitive diagnoses and treatment planning, as it can confirm the histopathological subtype and receptor status of the metastatic lesion [15]. Detecting metastases early can significantly affect treatment decisions, minimize morbidity, and improve survival outcomes.

Emerging imaging modalities, such as hyperspectral imaging (HSI) combined with computer-aided diagnostic (CAD) tools, are showing significant promise in enhancing breast cancer detection. HSI captures spectral data beyond conventional imaging, improving the sensitivity and specificity of cancer detection. When paired with CAD, HSI allows for detailed analysis and early diagnosis. A recent meta-analysis demonstrated that this combination achieved a pooled sensitivity of 78%, a specificity of 89%, and an accuracy of 87%, with particularly strong results from support vector machine models, which outperformed other machine learning techniques in terms of diagnostic performance [16].

The aims of this study were to retrospectively review a series of patients diagnosed with orbital metastases from breast carcinoma across three Italian orbital oncology centers, and to assess the clinical and histological characteristics, typical imaging features, metastasis latency, DRS, and the mortality rate of patients with these metastases.

## 2. Materials and Methods

A retrospective analysis of 32 patients with incisional biopsy-proven BC orbital metastases, who were referred to the Ocular Oncology Unit of Fondazione Policlinico Universitario A. Gemelli IRCCS, the Orbital Unit of Azienda Ospedaliera Universitaria Federico II of Naples, and the Orbital Service of Tiberia Hospital—GVM Care & Research over a period of 7 years from January 2016 to December 2023 was carried out.

Although most patients presented with an orbital metastasis following a previous history and diagnosis of primary BC, in three cases, the primary tumor was unknown at the time of the diagnosis of orbital metastasis.

The diagnosis of BC orbital metastasis was based on orbital incisional biopsy and subsequent histopathological examination. The patients underwent a complete ophthalmological investigation that included visual acuity assessment, tonometry, exophthalmometry, and ocular motility examination. To evaluate the extent and localization of orbital metastases, both CT scans and MRIs were performed. CT scans were obtained with axial and coronal reconstructions after intravenous contrast administration. MRI protocols included T1-weighted, T2-weighted, and fat-suppressed contrast-enhanced sequences in axial, coronal, and sagittal planes. Most cases were discussed in multidisciplinary ocular oncology boards to define the proper staging and treatment [7,8,17].

The main outcomes of the study were the analysis of the demographic characteristics of the population, the onset of ophthalmological clinical features, the imaging and histopathological features, the latency of orbital metastasis, the disease-related survival, and the mortality rate.

We have defined our survival endpoint as overall survival from the time of orbital metastasis diagnosis. To mitigate the inherent limitations of a retrospective multicenter design, the following measures were applied to enhance data quality and consistency: a standardized data collection form was used across centers, based on pre-defined clinical and pathological variables; data were centrally reviewed for completeness and consistency; discrepancies were resolved through direct queries with the contributing centers; missing data were quantified and handled conservatively, with key analyses restricted to variables with sufficient completeness; and pathological and imaging assessments were confirmed by senior specialists to ensure consistency with current standards.

The study was approved by the Institutional Ethics Committee of the Catholic University/Fondazione Policlinico Universitario A. Gemelli IRCCS (protocol ID number: 5896).

### Descriptive Statistics

To describe the distribution of demographic characteristics and clinical data in the study population, numerical variables are presented as median (interquartile range), and categorical variables are presented as count (percentage). For categorical variables, 95% confidence intervals for proportions were calculated using multinomial confidence intervals. Moreover, disease-related survival (DRS) was analyzed using Kaplan–Meier survival curves. All statistical analyses were performed using R software (version 4.3.1, https://www.R-project.org/).

Written informed consent for data collection and analysis for research purposes was obtained from all patients. The study was conducted in accordance with the Declaration of Helsinki. The patients were identified through our electronic databases based on the SKIN-COBRA system, which allows clinicians to retrospectively retrieve anonymous information on the patients themselves, consequently fully complying with the GDPR (General Data Protection Regulation) [18].

## 3. Results

Thirty-two patients affected by BC orbital metastasis and admitted to three Italian ophthalmological centers after referral were included.

Table 1 summarizes the main demographic and clinical characteristics of the population. The median age of the patients included in the study was 62.50 years (74.50–57.50), while the age of onset of primitive breast cancer was 55.50 years (61.25–45.00). Metastases to the left eye were observed in 17 cases (53.12%, 95% CI: 37.50–71.29%), while 14 cases involved the right eye (43.75%, 95% CI: 28.12–61.91%). Bilateral metastases were rare, occurring in only one case (3.12%, 95% CI: 0.00–21.29%). The confidence intervals for right and left eye metastases overlap substantially, suggesting no significant differences in laterality. With regard to breast cancer histotype, 23 (71.88%, 95% CI: 59.37–88.35%) patients had a lobular histotype, 8 (25.00%, 95% CI: 12.50–41.48%) had a ductal histotype, and only 1 (3.12%, 95% CI: 0.00–19.60%) patient presented a mixed histotype (Table 1). The confidence intervals indicate that the proportion of lobular cases is significantly higher compared to the other subtypes. The median orbital metastasis latency was 39.50 months (134.00–10.25) (Table 1).

Figure 1 shows the Kaplan–Meier curve of disease-related survival (DRS) in the study cohort. The median DRS was 35 months, and the curve plateaus at approximately 40% beyond 50 months, indicating that fewer than half of the patients survived beyond this time point. This early drop and sustained plateau suggest that a considerable proportion of disease-related deaths occurred during the early follow-up period. The 12-, 24-, and 36-month survival rates extracted from the Kaplan–Meier curve were, respectively, 86.44% (95% CI: 74.88–99.78%), 70.73% (95% CI: 55.43–90.25%), and 49.28% (95% CI:32.97–73.66%) (Figure 1).

The raw disease-related mortality data rate over the entire observation period (1 January 2016–31 December 2023) was 50.00% (Table 1).

Histological analysis revealed dense connective fibrous tissue in all cases with neoplastic infiltration by irregular nests or cords of epithelial cells with hyperchromatic, moderately atypical nuclei (altered N/C ratio, coarse chromatin, and indistinct nucleoli), and low–moderate amounts of amphiphilic cytoplasm. Neoplastic cells were positive for estrogen receptors, progesterone receptors, GATA3, and AE1/AE3. E-cadherin was negative. The HER2 receptor was found to be negative via immunohistochemistry and/or silver in situ hybridization (no amplification was detected) (Figure 2).

MRIs of all BC orbital metastases showed diffuse extra-bulbar fat infiltration by pathological tissue, heterogeneously hypointense on both T1w and T2w images, with marked contrast enhancement. The extraocular muscles were enlarged, and no signs of orbital bone infiltration were detected on CT scans in any of the cases.

Table 2 presents the clinical characteristics of our patients with orbital metastases from breast cancer, including gender, affected eye, and the histotype of the primary breast cancer. In most cases, the histotype of the primary tumor and orbital metastasis matched; however, two patients (cases 3 and 12) showed discordance, with lobular histology in the metastasis differing from the ductal/mixed subtype of the primary tumor. It also details the age at breast cancer detection, the histotype of the metastasis, and the latency period from primary cancer diagnosis to orbital metastasis onset. The table further includes clinical features, mortality rates, and DRS outcomes.

## 4. Discussion

The most typical localization of BC orbital metastases is the upper lateral quadrant of the extraconal space, which accounts for 50% of cases. The presentation in the intraconal space represents 30% of cases, and there may be a bilateral orbital involvement in up to 20% of cases [4,5,19]. The signs of the disease include conjunctival chemosis, the presence of a palpable mass, diplopia due to ocular motility impairment, proptosis [20], and optic nerve compression. In about 10% of cases, a retraction of the orbital tissues may determine the typical “scirrhous” presentation with enophthalmos [3].

In a retrospective study conducted by Grajales-Alvarez et al. [7] on 28 patients with orbital metastases from BC, the prevalent histotype was lobular carcinoma, accounting for 85% of cases. At the time of diagnosis of the metastasis, the disease was already at an advanced stage (stage IV), with an average age of 50.8 years. Most patients were positive for estrogen and progesterone receptors (Luminal A), one case was HER2/neu-positive (Luminal B), and one case was triple-negative.

In accordance with this data, most of the patients in our series had histology revealing an invasive lobular carcinoma (ILC) histotype, which is consistent with the primary tumor; however, a histological difference between the primary tumor and the orbital metastasis was observed in two cases. A similar finding was reported by Blohmer et al. [21], providing further evidence of the peculiar ILC spreading pattern. The high level of estrogen receptors and aromatase expression of estrogen and its production by mesenchymal cells of the orbital adipose tissue could explain the greater probability of invasive lobular carcinoma to metastasize to the orbit and other atypical sites, providing a favorable ground for BC ILC metastatic cells’ growth [22,23,24]. In our series, the orbital metastases were detected before the primary tumor in 3 out of 32 cases (9.37%; Table 2). According to a previous report, orbital metastases are diagnosed before the primary tumor in approximately 26% to 32%. Particularly, in a series of 93 cases, 30 patients had no history of cancer, and the primary tumor was subsequently located in 14 cases as follows: 6 in breast cancer, 3 in lung cancer, and 1 in kidney, prostate gland, adrenal gland, gastrointestinal tract, and liver cancers [1,5].

Orbital metastases occur in advanced stages of disease [2,25]. This is also supported by our data: the median disease-free survival (DFS) was 35 months (27.25–14.50 IQR); the 12-, 24-, and 36-month survival rates were, respectively, 86.44% (IQR: 74.88–99.78%), 70.73% (IQR: 55.43–90.25%), and 49.28% (IQR: 32.97–73.66%); the mortality rate was 50.00% during this observation period.

However, this raw mortality rate dataset does not account for censored cases—i.e., patients who were still alive at the end of the follow-up—potentially underestimating the true cumulative incidence of disease-related death. In this regard, Kaplan–Meier estimates provide a more accurate survival profile over time. This provides evidence of an advanced stage of disease when orbital metastases are primarily detected [8,9,10,11,12,13,14,15,16,17,18,19,20,21,22,23,24,25,26]. However, this could also be related to a more aggressive morphological and molecular modification of the metastasis when compared with the primary tumor [27,28,29].

Molecular features of the tumor have predictive and prognostic value and are relevant for treatment planning [10,11,27]. Stålhammar et al. [15] showed that more than two-thirds of the BC orbital metastases were found to be the hormone-receptor-positive Luminal B subtype, a rather aggressive form of cancer requiring generally cytotoxic chemotherapy [29]. In the same study, more than half of the patients showed positivity for Human Epidermal Growth Factor Receptor 2 (HER2), which would probably benefit from a therapy with monoclonal antibodies such as trastuzumab [27,28,29]. It has also been reported that the biomarker status may differ between the primary tumor and the orbital metastases, and biopsy of the metastatic lesion is recommended before treatment planning [15].

The ILC histologic subtype represents 10–15% of primary BC subtypes and shows a higher tendency for distant metastases than an IDC of the same grade [30,31]. It is generally associated with hormone receptor positivity (HR), low proliferative rates, good prognoses, and low histological grades [21] and represents the most frequently reported histological subtype of BC orbital metastases [22,23]. In a study by Blohmer et al., the survival rate of the patient after the diagnosis of ILC distant metastases was significantly shorter than for patients with invasive ductal carcinoma (IDC) [21]. This may suggest that when the tumor progresses to the stage of orbital metastases, it becomes histologically and clinically more aggressive [21].

In most cases of our series, the histology of the primary tumor and metastasis matched. Discordance was reported in two cases in which the primary tumor was a mixed ductal subtype and ductal subtype, while the orbital metastasis showed a lobular histotype.

We found a median orbital metastasis latency of 39.50 months (134.00–10.25). The literature reports an average latency time between the diagnosis of BC and the onset of orbital metastases between 2.0 and 8.5 years [32].

As for clinical features and disease presentation, exophthalmos was the most common sign observed in our case series (27 out of 32 cases), often with the involvement of extraocular muscles, with diplopia, as well as ophthalmoplegia (Table 2). Five out of thirty-two cases presented enophthalmos and the retraction of periorbital tissues. In a review of 25 cases with orbital metastases presenting with enophthalmos, Reifler et al. [33] reported that 76% were breast cancer metastases. The onset of progressive bilateral enophthalmos representing the first manifestation of metastatic BC was also described by Gonçalves et al. [34] in a case report.

On MRIs, in all cases, BC orbital metastases presented diffuse extra-bulbar fat infiltration by pathologic tissue, being heterogeneously hypointense on both T1w and T2w images, with marked contrast enhancement. The extraocular muscles were usually swollen. Scirrhous BC metastases in the involved eye characteristically presented enophthalmos (Figure 3), probably due to infiltration and desmoplasia, while in non-scirrhous metastases, exophthalmos was most frequently encountered (Figure 4).

In one case, imaging showed thickening of the right medial rectus muscle, and no other abnormalities (case 13).

No significant differences were observed on histopathological examination and MRI signal and enhancement between enophthalmos and exophthalmos cases. More frequently, the scirrhous type presented hypointense components on T2w images, possibly related to a desmoplastic and fibrotic reaction, which may eventually be detected with ultrasound [35,36,37].

Pathologic intra-orbital tissue is characterized by hypointense signals on T2w sequences and marked contrast enhancement, diffusely involving the intraconic and extraconic right orbital fat. The involved extrinsic ocular muscles are slightly thickened; enophthalmos of the right eye is evident.

The pathologic tissue is characterized by inhomogeneous low signals on both T1w and T2w sequences and marked contrast enhancement diffusely involving intraconic and extraconic orbital fat. The involved extrinsic ocular muscles are markedly thickened; exophthalmos of the right eye is evident.

## 5. Conclusions

The herein reported case series further supports the idea that lobular carcinoma accounts for most orbital metastases, although the prevailing primary tumor is the ductal histotype (no special type—NST), suggesting a preferential trophism of the lobular carcinoma subtype to the orbital region. In contrast to what is reported in the literature, in our series, there were only 2 out of 32 cases presenting histological discordance between the primary tumor and the orbital metastasis; in all the other cases, the histological profile of the metastasis was concordant with the primary tumor. The clinical presentation of the diseases was consistent with those of previous studies, and the most frequently observed signs were exophthalmos, diplopia, and the restriction of ocular movements, or alternatively, enophthalmos and orbital tissue retraction. Increasing mortality and DFS rates are found in advanced stages of the disease, along with a transformation into a more aggressive histotype. Diffuse orbital fat involvement, characterized by hypointensity on both T1w and T2w sequences, as well as marked contrast enhancement, was the most common MRI feature.

As treatments for primary BC have improved, leading to increased survival rates, the incidence of orbital metastasis is expected to rise. This underscores the importance of long-term surveillance and the need to remain vigilant for symptoms indicative of orbital involvement in BC survivors. Ophthalmological evaluation should be performed on all patients with known BC and mild ocular symptoms. Early local treatment may help to prevent severe complications and reduce negative impacts on patients’ quality of life [38].

As a retrospective study, our analysis is subject to inherent limitations, including variability in clinical documentation and diagnostic assessments across centers, and stratified analyses based on systemic treatments, radiation, or surgical interventions were not performed. We do acknowledge the potential for referral bias, which may limit the generalizability of our findings.

Although these factors may influence the accuracy of the study, and a prospective design would indeed offer greater control, we believe that the rigorous data harmonization steps applied to our study enhance the reliability of our findings. Moreover, we are currently exploring opportunities for collaboration with additional centers to extend the dataset and assess the reproducibility of our results in a larger cohort.

## Figures and Tables

**Figure 1 cancers-17-01875-f001:**
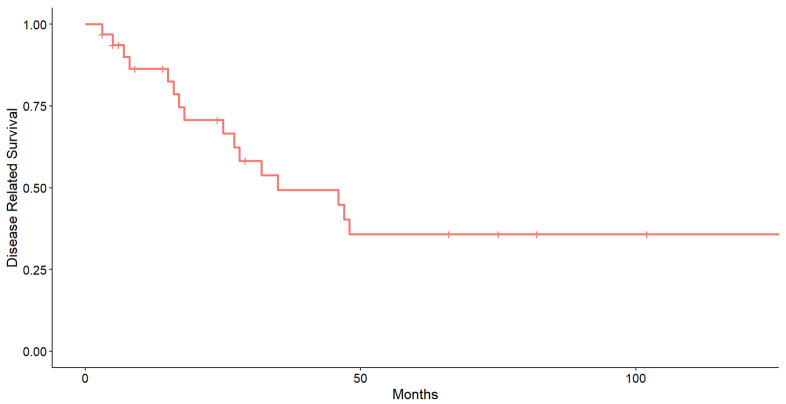
Kaplan–Meier survival curve of disease-related survival (DRS).

**Figure 2 cancers-17-01875-f002:**
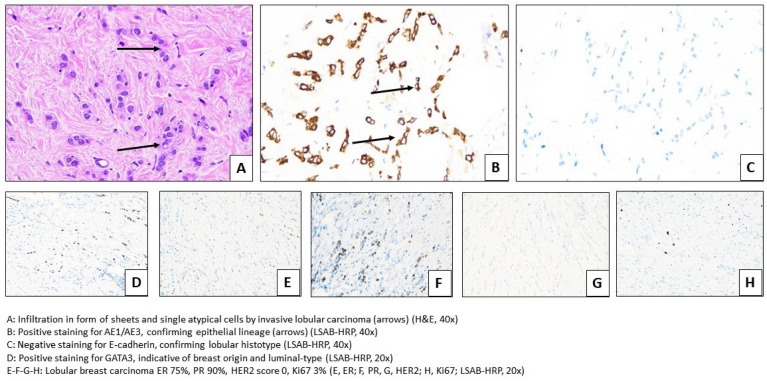
Histopathological and immunohistochemical features.

**Figure 3 cancers-17-01875-f003:**
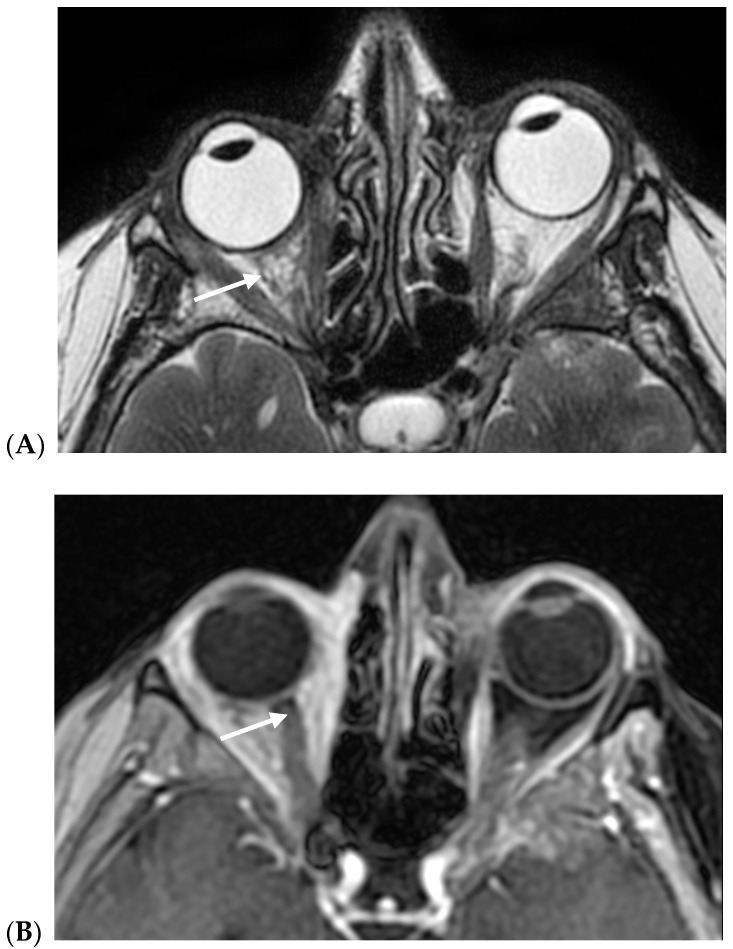
(**A**,**B**) Right orbital scirrhous breast cancer metastasis with enophthalmos (arrows): (**A**) axial fast spin echo of T2w images; (**B**) axial fast spin echo contrast-enhanced fat-sat T1w images.

**Figure 4 cancers-17-01875-f004:**
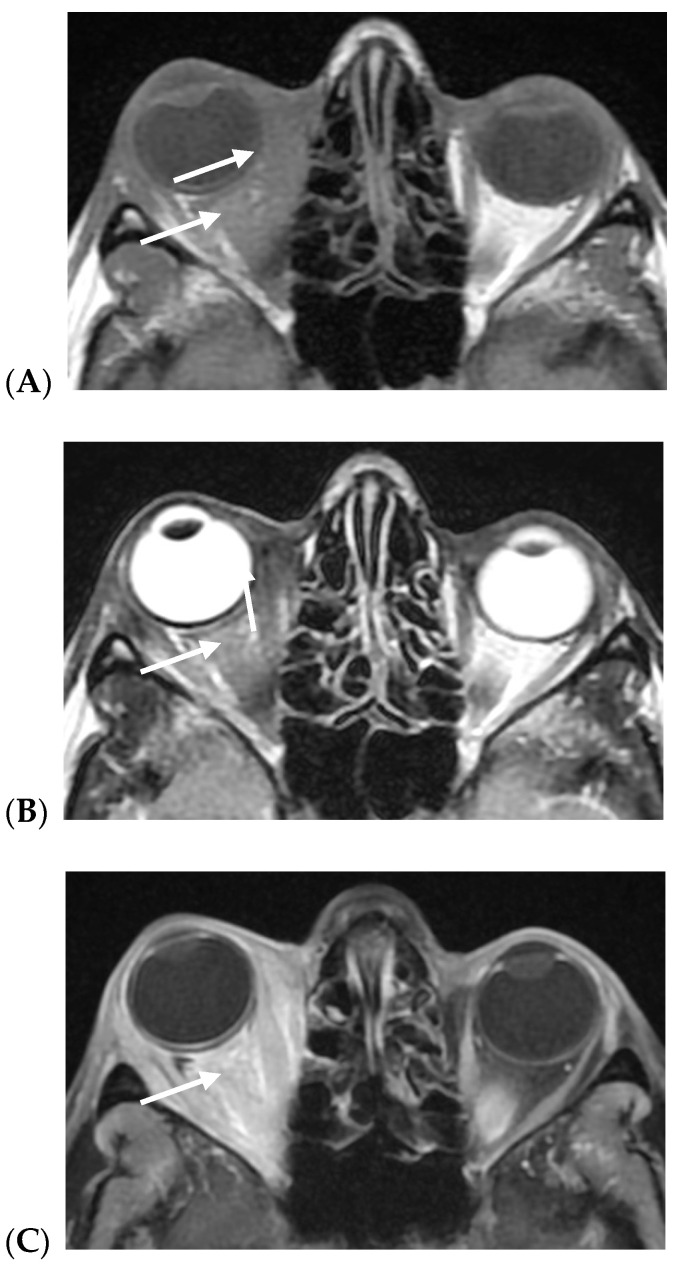
(**A**–**C**) Right orbital non-scirrhous breast cancer with exophthalmos (arrows): (**A**) axial fast spin echo of T2w and (**B**) T1w images; (**C**) contrast-enhanced fat-sat T1w images.

**Table 1 cancers-17-01875-t001:** Demographic characteristics and clinical data of the population. Numerical variables are presented as median (IQR), and categorical variables are presented as count (percentage).

		Overall Population (N = 32)
**Age (years)**		62.50 (74.50–57.50)
Eye	RE	14.00 (43.75%, 95% CI: 28.12–61.91%)
	LE	17.00 (53.12%, 95% CI: 37.50–71.29%)
	Bilateral	1.00 (3.12%, 95% CI: 0.00–21.29%)
**Age at onset of primitive breast cancer** (years)		55.50 (61.25–45.00)
Breast cancer histotype	Lobular	23.00 (71.88%, 95% CI: 59.37–88.35%)
	Mixed	1.00 (3.12%, 95% CI: 0.00–19.60%)
	Ductal	8.00 (25.00%, 95% CI: 12.50–41.48%)
Orbital metastasis latency (months)		39.50 (134.00–10.25)
Mortality rate	Alive	16.00 (50.00%)
	Dead	16.00 (50.00%)

**Table 2 cancers-17-01875-t002:** Clinical cases. RE: right eye; LE: left eye.

	Gender	Eye	Primitive Breast Cancer Histotype	Primitive Breast Cancer Age (Years)	Orbital Metastasis Hystotype (Breast Biopsy)	Latency Onset Months	Exopthalmos/	Mortality	Disease-Related Survival (Months)	Other Clinical Presentations
							Enopthalmos			
Case 1	F	RE	Lobular	49	Lobular	5	Exophthalmos	Dead	28	
Case 2	F	Bilateral	Lobular	65	Lobular	142	Exophthalmos	Dead	18	Diplopia, eyelid edema, ophthalmoplegia
Case 3	F	RE	Mixed	67.5	Lobular	5	Enophthalmos	Alive	29	Ptosis, ophthalmoplegia
Case 4	F	RE	Lobular	49	Lobular	10	Exophthalmos	Dead	25	Complete ophthalmoplegia
Case 5	F	RE	Lobular	53	Lobular	n/a	Enophthalmos	Alive	9	
Case 6	F	RE	Lobular	77	Lobular	5	Exophthalmos	Dead	32	
Case 7	F	RE	Lobular	66.5	Lobular	5	Exophthalmos	Dead	16	
Case 8	F	RE	Lobular	60	Lobular	11	Exophthalmos	Alive	14	Periorbital edema
Case 9	F	LE	Lobular	60	Lobular	9	Exophthalmos	Dead	17	Funnel retinal detachment with multiple choroidal lesions
Case 10	F	LE	Lobular	59	Lobular	53	Enophthalmos	Alive	3	Ptosis
Case 11	F	RE	Lobular	56	Lobular	348	Enophthalmos	Alive	6	
Case 12	F	LE	Ductal	62	Lobular	n/a	Exophthalmos	Alive	6	
Case 13	F	RE	Lobular	40	Lobular	192	Enophthalmos	Dead	8	Enlarged right medial rectus
Case 14	F	LE	Lobular	45	Lobular	138	Exophthalmos	Dead	15	UE hard swelling
Case 15	F	LE	Lobular	48	Lobular	220	Exophthalmos	Alive	24	Ptosis, complete ophthalmoplegia
Case 16	F	RE	Ductal	78	Ductal	76	Exophthalmos	Dead	47	Exotropia
Case 17	F	LE	Ductal	44	Ductal	154	Exophthalmos	Alive	66	Ptosis, diplopia, UE hard swelling
Case 18	F	LE	Ductal	55	Ductal	26	Exophthalmos	Alive	146	-
Case 19	F	LE	Lobular	32	Lobular	106	Exophthalmos	Alive	102	Ptosis, diplopia
Case 20	F	LE	Lobular	75	Lobular	33	Exophthalmos	Dead	27	Complete ophthalmoplegia
Case 21	F	RE	Lobular	61	Lobular	19	Exophthalmos	Alive	165	-
Case 22	F	LE	Ductal	35	Ductal	14	Exophthalmos	Dead	5	Diplopia
Case 23	F	LE	Lobular	56	Lobular	25	Exophthalmos	Alive	195	Ptosis
Case 24	F	RE	Ductal	35	Ductal	166	Exophthalmos	Dead	7	Ptosis, diplopia, UE swelling
Case 25	F	LE	Lobular	45	Lobular	37	Exophthalmos	Alive	75	Ptosis
Case 26	F	LE	Ductal	53	Ductal	94	Exophthalmos	Dead	35	Diplopia
Case 27	F	LE	Ductal	35	Ductal	93	Exophthalmos	Alive	211	Ptosis, diplopia
Case 28	F	LE	Lobular	42	Lobular	n\a	Exophthalmos	Dead	3	Diplopia, complete ophthalmoplegia
Case 29	F	RE	Lobular	58	Lobular	324	Exophthalmos	Alive	5	Ptosis, exo-ipotropia
Case 30	F	LE	Lobular	48	Lobular	122	Exophthalmos	Dead	46	Hard swelling UE
Case 31	F	RE	Lobular	77	Lobular	1	Exophthalmos	Dead	48	Complete ophthalmoplegia
Case 32	F	LE	Lobular	60	Lobular	42	Exophthalmos	Alive	82	Ptosis

## Data Availability

The data that support the findings of this study are available from G.C., upon reasonable request.

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
