# Peer review of "Breast Cancer Orbital Metastases: Clinical and Histopathological Characteristics, Imaging Features, and Disease-Related Survival in a Multicentric Retrospective Case Series"

_cancers, 2025, doi:10.3390/cancers17111875_

Round 1

Reviewer 1 Report

Comments and Suggestions for Authors

This study is a cohort of breast cancer cases with orbital metastasis. Orbital metastasis is a rare condition, and the concept of this study is meaningful.

  1. A critical point of the study is the follow-up period.
  2. Regarding the IHC image of E-cadherin: if the authors aim to demonstrate lobular carcinoma, it is recommended to include a full IHC panel.

Author Response

This study is a cohort of breast cancer cases with orbital metastasis. Orbital metastasis is a rare condition, and the concept of this study is meaningful.

A critical point of the study is the follow-up period.

Thank you for your insightful comment. We agree that the follow-up period is a critical aspect in assessing the validity and interpretability of survival outcomes. In our study, we report both the overall mortality rate (50.0%) and the Kaplan-Meier survival estimates, including the median disease-related survival (35 months) and survival probabilities at specific timepoints (12, 24, and 36 months). These timepoints were selected to reflect clinically meaningful intervals within the available follow-up period. We recognize that raw mortality rates can be influenced by varying lengths of follow-up among patients. Therefore, we utilized Kaplan-Meier analysis to provide a more accurate and time-adjusted estimate of survival, as it accounts for censored cases and differential follow-up durations. To address this point, we have clarified the length and completeness of follow-up in the Methods section and elaborated in the Discussion (lines 202–209) on how the Kaplan-Meier approach mitigates limitations associated with incomplete follow-up. We also now discuss the implications of our follow-up duration for interpreting long-term outcomes. Please see below our reply to Reviewer 3.

Regarding the IHC image of E-cadherin: if the authors aim to demonstrate lobular carcinoma, it is recommended to include a full IHC panel.

As suggested, Figure 2 has been modified, now enriched of the representative stainings of GATA3, ER, PR, HER2 and Ki67.

Reviewer 2 Report

Comments and Suggestions for Authors

Congratulations to the authors of this interesting work, who present a multicenter retrospective case study of medical records of 32 patients with orbital metastases of breast cancer.

Structure of the paper: the images presented in the "Discussion" section and Table 2 are results of the study and fit better in the "Results" chapter.

Abstract, line 63: The abbreviation "DRS" (disease related survival) is only spelled out the second time it appears in the text (line 63 in the Introduction chapter). Please explain such abbreviations the first time they appear in the text, or avoid to use the abbrevation.

Discussion, line 140 - 144: Avoid repeating results exactly in the "Discussion" chapter. These data have already been presented in the same form in the "Results" chapter.

The study is methodologically sound, but the structure of the paper should be revised, particularly in view of the above-mentioned objections.

Author Response

Congratulations to the authors of this interesting work, who present a multicenter retrospective case study of medical records of 32 patients with orbital metastases of breast cancer.

Structure of the paper: the images presented in the "Discussion" section and Table 2 are results of the study and fit better in the "Results" chapter.

Thank you for this suggestion, now modified as requested.

Abstract, line 63: The abbreviation "DRS" (disease related survival) is only spelled out the second time it appears in the text (line 63 in the Introduction chapter). Please explain such abbreviations the first time they appear in the text, or avoid to use the abbrevation.

Modified as requested.

Discussion, line 140 - 144: Avoid repeating results exactly in the "Discussion" chapter. These data have already been presented in the same form in the "Results" chapter.

Modified as requested.

The study is methodologically sound, but the structure of the paper should be revised, particularly in view of the above-mentioned objections.

Thank you for your observations, the paper has been modified accordingly.

Reviewer 3 Report

Comments and Suggestions for Authors

Authors evaluate some characteristics of the orbital metastasis from the breast cancer.
The contribution finds its place among the available results, motivates the importance for screening against the orbital metastasis i BC patients, and delivers important experimental record which is worthty of publication.
However some issues can be improved.
1. Most of the data is presented by median and IQR, but it is not possible for fractions. It would be reasonable to estimate some confidence interval for fractions as well (e.g. to see whether the metastasis to left or right eye is equiprobable or not).
2. Considering the above left-right preferences, maybe it would be reasonable to specify which breast was affected in subjects.
3. The Kaplan-Meier curve is barely commented. It may come surprising at first glance that the flat region is lower than 50% measured for the overall sample. This should be commented.
4. Relating to 3. - the overall death percentage may not be the best indicator to measure, because some of the subjects under investigation were sick too short before the trial ended (i.e. if the trial would be longer, they would probably die due to the disease, probably approaching the Kaplan-Meier result).
5. In line 105 there is a survival rate for variable time length - which includes IQR. How this was established? A sliding time window within the sample? This should be specified.
6. In the images it would be useful to provide some arrow that points to the affected regions of interest.

Author Response

Authors evaluate some characteristics of the orbital metastasis from the breast cancer.
The contribution finds its place among the available results, motivates the importance for screening against the orbital metastasis i BC patients, and delivers important experimental record which is worthty of publication. However some issues can be improved.

  1. Most of the data is presented by median and IQR, but it is not possible for fractions. It would be reasonable to estimate some confidence interval for fractions as well (e.g. to see whether the metastasis to left or right eye is equiprobable or not).

Thank you for your comment. We have now calculated and included 95% confidence intervals for all categorical proportions using multinomial distribution.

2.Considering the above left-right preferences, maybe it would be reasonable to specify which breast was affected in subjects.

Unfortunately, although we got the histopathological report, we do not have the data of the laterality of the breast initially involved for each patient. For this reason, we are unable to correlate the laterality of the orbit and breast involved.

  1. The Kaplan-Meier curve is barely commented. It may come surprising at first glance that the flat region is lower than 50% measured for the overall sample. This should be commented.

Thank you for pointing it out. We have now added a more detailed description in the Results section to address this point (lines 133-137).

  1. Relating to 3. - the overall death percentage may not be the best indicator to measure, because some of the subjects under investigation were sick too short before the trial ended (i.e. if the trial would be longer, they would probably die due to the disease, probably approaching the Kaplan-Meier result).

Thank you for your comment. In the manuscript, we report both the overall mortality rate (50.0%) and the Kaplan-Meier estimates, including the median disease-related survival (35 months) and survival probabilities at specific timepoints (12, 24, and 36 months). We agree that the Kaplan-Meier method provides a more accurate estimate of survival over time, as it properly accounts for censored cases and we have clarified this distinction in the Discussion section (lines 202-209).

  1. In line 105 there is a survival rate for variable time length - which includes IQR. How this was established? A sliding time window within the sample? This should be specified.

The survival rates at 12, 24, and 36 months were directly extracted from the estimated Kaplan-Meier survival curve. The corresponding confidence intervals reported alongside each time point are based on the Greenwood standard error estimate at those specific time points. To avoid confusion, we have revised the manuscript to refer to these intervals as 95% confidence intervals (95% CI) instead of interquartile range (IQR).

  1. In the images, it would be useful to provide some arrow that points to the affected regions of interest.

Arrows have been applied to the affected regions of figures, as suggested.

Reviewer 4 Report

Comments and Suggestions for Authors

The retrospective design of the study naturally includes biases, including selection bias, missing data, and inconsistencies in clinical record among centers.  How can the authors mitigate any discrepancies and resolve absent data in this multicentric retrospective dataset?  Have any data validation or harmonization methodologies been implemented?  A prospective study design would enhance data collection control, standardize imaging and histological assessment, and potentially mitigate bias.

 Despite the inclusion of 32 patients, this sample size is relatively small for a multicentric investigation conducted across seven years, potentially constraining the statistical power and generalizability of the results.  Do the authors intend to augment the study cohort or corroborate findings with a bigger or external dataset to enhance the conclusions?

 How did the authors address variations in systemic treatments, radiation, or surgical interventions among patients?

 Have the authors contemplated a comparative analysis to evaluate the impact of orbital metastases on survival in relation to other metastatic patterns?

 How can the authors mitigate the potential for referral bias, and would this constrain the generalizability of their findings to other clinical contexts?

The introduction section fails to sufficiently explain the clinical value of early breast cancer identification and does not reference recent advancements and research highlighting the importance of monitoring for metastatic spread.  Enhance the introduction by examining the contemporary landscape of breast cancer detection, referencing significant influential studies from the past five years such as:

1) Leung, Joseph-Hang, Riya Karmakar, Arvind Mukundan, Pacharasak Thongsit, Meei-Maan Chen, Wen-Yen Chang, and Hsiang-Chen Wang. "Systematic meta-analysis of computer-aided detection of breast cancer using hyperspectral imaging." Bioengineering 11, no. 11 (2024): 1060.

 Explicitly delineating survival endpoints is essential for interpretation and for the comparison of outcomes across studies.

Author Response

The retrospective design of the study naturally includes biases, including selection bias, missing data, and inconsistencies in clinical record among centers.  How can the authors mitigate any discrepancies and resolve absent data in this multicentric retrospective dataset?  Have any data validation or harmonization methodologies been implemented?  A prospective study design would enhance data collection control, standardize imaging and histological assessment, and potentially mitigate bias.

We thank the reviewer for this important observation. We acknowledge the inherent limitations of a retrospective multicenter design. To mitigate these, we applied several measures to enhance data quality and consistency:

- A standardized data collection form was used across centers, based on pre-defined clinical and pathological variables.

- Data were centrally reviewed for completeness and consistency; discrepancies were resolved through direct queries with the contributing centers.

- Missing data were quantified and handled conservatively, with key analyses restricted to variables with sufficient completeness.

- Where feasible, pathological and imaging assessments were confirmed by senior specialists to ensure consistency with current standards.

While a prospective design would indeed offer greater control, we believe that the rigorous data harmonization steps applied here enhance the reliability of our findings. These efforts are now described in the Methods and discussed as a study limitation (Coinclusions).

Despite the inclusion of 32 patients, this sample size is relatively small for a multicentric investigation conducted across seven years, potentially constraining the statistical power and generalizability of the results.  Do the authors intend to augment the study cohort or corroborate findings with a bigger or external dataset to enhance the conclusions?  

We appreciate the reviewer’s observation regarding sample size. We acknowledge that the limited cohort (n = 32) may constrain the statistical power and generalizability of the results. However, this reflects the rarity and highly selective nature of the clinical scenario studied, even across multiple centers over several years. We agree that expanding the cohort or validating the findings in an external dataset would strengthen the conclusions. As such, we are currently exploring opportunities for collaboration with additional centers to extend the dataset and assess the reproducibility of our results in a larger, independent cohort. This limitation has been acknowledged in the Conclusions section, and we have included a statement regarding the need for future validation efforts.

How did the authors address variations in systemic treatments, radiation, or surgical interventions among patients?

Due to the retrospective nature and sample size of our study, we did not perform stratified analyses based on systemic treatments, radiation, or surgical interventions. However, we have included treatment details in the manuscript and now acknowledge this variability as a limitation in the conclusion section.

Have the authors contemplated a comparative analysis to evaluate the impact of orbital metastases on survival in relation to other metastatic patterns?

We appreciate this suggestion. While a comparative survival analysis between orbital and other metastatic patterns would be informative, it was beyond the scope of our current study due to limitations in sample size and available comparative data. However, we agree this is an important area for future research.

How can the authors mitigate the potential for referral bias, and would this constrain the generalizability of their findings to other clinical contexts?

Thank you for this comment. We acknowledge the potential for referral bias, which may limit the generalizability of our findings. To address this, we have clearly described our cohort and inclusion criteria, and where possible, conducted subgroup analyses to test the robustness of our results. We have also added a statement in the conclusions acknowledging this limitation.

The introduction section fails to sufficiently explain the clinical value of early breast cancer identification and does not reference recent advancements and research highlighting the importance of monitoring for metastatic spread.  Enhance the introduction by examining the contemporary landscape of breast cancer detection, referencing significant influential studies from the past five years such as:

Leung, Joseph-Hang, Riya Karmakar, Arvind Mukundan, Pacharasak Thongsit, Meei-Maan Chen, Wen-Yen Chang, and Hsiang-Chen Wang. "Systematic meta-analysis of computer-aided detection of breast cancer using hyperspectral imaging." Bioengineering 11, no. 11 (2024): 1060.

The introduction has been modified as requested and the suggested reference is now included.

Explicitly delineating survival endpoints is essential for interpretation and for the comparison of outcomes across studies.

Thank you for the comment. We have now clearly defined our survival endpoint as overall survival from the time of orbital metastasis diagnosis, and have clarified this in the Methods section.

Reviewer 5 Report

Comments and Suggestions for Authors

Thank you for the opportunity to review this interesting retrospective multicentric study titled "Breast Cancer Orbital Metastases: Clinical, Imaging, Histopathological Features, and Disease-Related Survival in a Multicentric Retrospective Case Series."

However, some improvements in the methods and results are necessary as there are poor descriptions in the results of the imaging findings, and the manuscript should be reordered as some results information was reported in the discussion.

In the methods, authors should specify the imaging modality used to study orbital metastasis. If only MRI was used, authors should specify the protocols. If both MRI and CT were used, authors should specify the protocols.

In the results, authors should specify some features of the primitive breast cancers, such as the TNM stage, if multifocal, multicentric, and the breast quadrant involved.

Authors reported some results with histologic findings and MRI images in the discussion. However, they should be reported in the results and discussed afterward.

Authors should report the discordant cases in the results. Lines 166-176 should be reported in the results, and these findings should be discussed afterward.

Authors should also specify in the results the imaging features of orbital metastasis on MRI, including the morphology, signal, and localizations.

In lines 205-206, what case showed only the thickening of the right medial rectus muscle? Authors should specify this in the results and discuss this finding in the discussion.

In lines 137-138, the orbital metastases were detected before the primary tumor in 3 out of 32 cases (9.37% - Table 2). Authors should specify which cases these are.

In the table 2 the authors should specify the acronyms used: LE, RE

Author Response

Thank you for the opportunity to review this interesting retrospective multicentric study titled "Breast Cancer Orbital Metastases: Clinical, Imaging, Histopathological Features, and Disease-Related Survival in a Multicentric Retrospective Case Series."

However, some improvements in the methods and results are necessary as there are poor descriptions in the results of the imaging findings, and the manuscript should be reordered as some results information was reported in the discussion.

In the methods, authors should specify the imaging modality used to study orbital metastasis. If only MRI was used, authors should specify the protocols. If both MRI and CT were used, authors should specify the protocols.

Thank you for your comment. We have clarified in the manuscript that both CT and MRI have been used as imaging modalities in the evaluation of orbital metastases, and we have specified the imaging protocols used for each technique.

In the results, authors should specify some features of the primitive breast cancers, such as the TNM stage, if multifocal, multicentric, and the breast quadrant involved.

Unfortunately, although we got the histopathological report, we do not have all this information in our dataset.

Authors reported some results with histologic findings and MRI images in the discussion. However, they should be reported in the results and discussed afterward.

Thank you for the comment. We have included both the histological findings and the MRI imaging features in the Results, where they are now appropriately described.

Authors should report the discordant cases in the results. Lines 166-176 should be reported in the results, and these findings should be discussed afterward.

Thank you for your suggestion. We have included the discordant cases in the Results  and discussed them in the Discussion.

Authors should also specify in the results the imaging features of orbital metastasis on MRI, including the morphology, signal, and localizations.

Thank you for pointing this out. This information has been added to the Results.

In lines 205-206, what case showed only the thickening of the right medial rectus muscle? Authors should specify this in the results and discuss this finding in the discussion.

It was case 13, information added as requested.

In lines 137-138, the orbital metastases were detected before the primary tumor in 3 out of 32 cases (9.37% - Table 2). Authors should specify which cases these are.

In our series, orbital metastases were detected before the diagnosis of the primary tumor in 3 out of 32 cases (9.37%; cases 3, 5, and 12; Table 2), which are identified in the table by “Latency onset months: N/A.”

In the table 2 the authors should specify the acronyms used: LE, RE.

Modified as requested

Round 2

Reviewer 2 Report

Comments and Suggestions for Authors

Thank you for the consistent implementation of the proposed changes during the last peer review!

Reviewer 4 Report

Comments and Suggestions for Authors

All the comments has been addressed

Reviewer 5 Report

Comments and Suggestions for Authors

Thank you for the requested modifications that have improved the quality of this interesting manuscript. Congratulations